# Biofortification of Pulse Crops: Status and Future Perspectives

**DOI:** 10.3390/plants9010073

**Published:** 2020-01-06

**Authors:** Ambuj B. Jha, Thomas D. Warkentin

**Affiliations:** Crop Development Centre/Department of Plant Sciences, University of Saskatchewan, 51 Campus Drive, Saskatoon, SK S7N 5A8, Canada; ambuj.jha@usask.ca

**Keywords:** biofortification, iron, zinc, selenium, iodine, carotenoid, folate, pulse

## Abstract

Biofortification through plant breeding is a sustainable approach to improve the nutritional profile of food crops. The majority of the world’s population depends on staple food crops; however, most are low in key micronutrients. Biofortification to improve the nutritional profile of pulse crops has increased importance in many breeding programs in the past decade. The key micronutrients targeted have been iron, zinc, selenium, iodine, carotenoids, and folates. In recent years, several biofortified pulse crops including common beans and lentils have been released by HarvestPlus with global partners in developing countries, which has helped in overcoming micronutrient deficiency in the target population. This review will focus on recent research advances and future strategies for the biofortification of pulse crops.

## 1. Introduction

Micronutrients, iron (Fe), zinc (Zn), selenium (Se), iodine (I), carotenoids, and folates are essential nutrients required for human growth and development, as these contribute to various metabolic functions in human. The majority of the world’s population depends on plant-based foods which are often low in key micronutrients [1], and do not meet the recommended daily allowances (RDA). Micronutrient malnutrition is commonly known as “hidden hunger” and affects one in three people worldwide [2]. Micronutrient deficiencies may lead to serious illnesses such as poor growth, intellectual impairments, perinatal complications, and increased risk of morbidity and mortality [3]. Further, they aggravate infectious and chronic diseases including osteoporosis osteomalacia, thyroid deficiency, colorectal cancer, and cardiovascular diseases and thus greatly impact quality of life [4].

Deficiencies of Fe, Zn, folic acid, and β-carotene are global issues, but they are more predominant in Asian, African, and Latin American countries and affect more than two billion people [4,5]. Micronutrient deficiency and undernourishment of pregnant mothers affects nearly 50% of the world’s population, potentially leading to intrauterine growth restriction, low birth weight, protein-energy malnutrition, and chronic energy deficit [6]. Though rates are higher in Africa and Asia, deficiencies of the four common micronutrients, Fe, I, Zn, and vitamin A, alone are responsible for about 12% of deaths globally among children under 5 years of age [6].

Food crops rich in nutrients could address deficiencies of micronutrients and thus provide a sustainable solution to global health issues [7]. Peas (*Pisum sativum* L.), chickpeas (*Cicer arietinum* L.), lentils (*Lens culinaris* Medik.), common beans (*Phaseolus vulgaris* L.), and mungbeans (*Vigna radiate* L.) are major pulse crops grown worldwide [8]. They are great sources of dietary proteins, complex carbohydrates, vitamins, and minerals required for human nutrition [9,10,11,12,13,14,15]. Pulse crops are used in traditional diets of people in many parts of the world since they are rich in proteins and amino acid and are slowly digestible carbohydrates [8,9,16]. They are easily available to all groups of people on a regular basis and provide the least expensive source of proteins and micronutrients [17]. Pulse consumption has been increasing owing to their health and environmental benefits [18].

Micronutrient malnutrition has received increased attention in recent decades at a global level and efforts have been made to combat them by various strategies such as increased food production, supplementation, food fortification, and biofortification. Biofortification, enriching the nutritional quality of food crop using either conventional plant breeding or modern biotechnology, is a balanced approach to overcome mineral deficiencies [19,20,21,22]. Biofortification through plant breeding to improve the nutritional profile of pulse crops has gained momentum in the past decade. In this regard, several studies in pulse crops have identified genetic variation for the key micronutrients in the available gene pools, with promising breeding lines being used in breeding, and associated genotypic markers for marker assisted selection [11,12,13,23,24,25,26,27,28,29,30]. This review will focus on recent research advances for the improvement of key micronutrients, Fe, Zn, Se, I, carotenoids, and folates in pulse crops. This review will also discuss challenges and future strategies for the biofortification of pulse crops.

## 2. Key Micronutrients

### 2.1. Iron

Iron (Fe) is indispensable for living organisms and vital for various metabolic processes such as electron transport and deoxyribonucleic acid synthesis [31]. In the human body, Fe is required for the synthesis of oxygen transport proteins (hemoglobin and myoglobin) and enzymes involved in electron transfer and oxidation-reductions [32,33]. In hemoglobin, it serves as a transporter of oxygen from the lungs to the tissues. According to the Food and Nutrition Board of the Institute of Medicine, National Academy of Sciences, the RDA of Fe is 8 mg/day for adult males and 18 mg/day for females (https://ods.od.nih.gov/Health_Information/Dietary_Reference_Intakes.aspx). Iron deficiency is considered the most predominant among various micronutrient deficiencies and the major contributor of anemia and affects more than two billion individuals globally [34,35]. It can cause loss of energy, dizziness, and poor pregnancy outcomes such as premature births, low birth weight babies, delayed growth and development in infants, and poor cognitive skills [3,36,37].

### 2.2. Zinc

Zn is an another important mineral required by humans and is involved in many biological functions, such as improving wound healing by its involvement in membrane signaling systems in cell growth and proliferation [38,39], protecting cells from oxidative damage by quenching reactive oxygen species [40,41], and reducing risk of various cancers including prostate and pancreatic [42]. The RDA for Zn is 11 mg/day for adult males and 8 mg/day for adult females (https://ods.od.nih.gov/Health_Information/Dietary_Reference_Intakes.aspx). Deficiency of Zn has many consequences including a weak immune system, recurrent infections, mental illness, and retarded growth and fertility [43]. It plays an important role in cell division; it thus significantly affects pregnant women.

### 2.3. Selenium

Se, an essential micronutrient, is required for growth and development and protects the human body against infection, oxidative stress, and progression of cancer [44,45,46,47]. The RDI for Se is 55 μg/day for both males and females (https://ods.od.nih.gov/Health_Information/Dietary_Reference_Intakes.aspx). In humans, Se deficiency is associated with several diseases, such as Keshan, Keshin-Beck, and myxedematous cretinism [48].

### 2.4. Iodine

Iodine is a vital constituent of the thyroid hormones, thyroxine (T4), and triiodothyronine (T3) and essential for normal growth, development, and metabolism. According to the Food and Nutrition Board, Institute of Medicine, the RDI for I is 150 μg/day for both adult males and females (https://ods.od.nih.gov/Health_Information/Dietary_Reference_Intakes.aspx). Deficiency of iodine causes hypothyroidism, goiter, cretinism, mental retardation, and reduced fertility and is accountable for increased prenatal death and infant mortality [49,50,51]. Deficiency during pregnancy can cause cognitive impairment in the offspring as it is critical for brain development [52,53].

Deficiency of iodine in human populations is different from other micronutrients as it is predominant in developing as well as developed countries [54,55,56]. This could be due to the low concentration of this mineral in agricultural soils and cereal-based foods [55].

### 2.5. Carotenoids

Carotenoids are natural pigments produced by plants. Plant-derived foods are sources of carotenoids as humans and animals cannot synthesize carotenoids [57]. Carotenoids act as important antioxidants in the human body and play a key role in various physiological processes. Overall, more than 600 carotenoids are known. Lutein and zeaxanthin prevent age-related macular degeneration [57,58]. Lutein reduces the risk of cataracts and is associated with cardiovascular disease prevention [59,60]. Vitamin A is important for normal vision, bone growth, and cell division in mammals [61]. β-Cryptoxanthin stimulates osteoblastic bone formation and inhibits osteoclastic bone resorption [62], therefore playing an important role in bone formation. Carotenoids have strong cancer-fighting properties [63] and protect cellular organelles from oxidative damage by efficiently scavenging free radicals generated during various metabolic processes [64,65]. Carotenoids are considered as Fe absorption promoters as these improve human Fe bioavailability from plant-based foods [7]. For example, improvement of Fe status was reported in the Venezuelan population after the addition of vitamin A in their food [66,67].

### 2.6. Folates

Folates are B9 vitamins and act as cofactors in various metabolic functions such as nucleotide biosynthesis and amino acid metabolism in the human body [68,69], and are therefore required for human growth and development. In plants, folates are important for biosynthesis of biomolecules including lignin, alkaloids, and chlorophyll [70]. Humans are dependent on plant and/or animal-based food sources as they cannot synthesize folates [69,71]. Deficiency of folates has been associated with greater risk of various chronic diseases, such as neural tube defects [72], impaired cognitive function [73], Alzheimer’s disease [74], cardiovascular diseases [75], and certain types of cancers [76]. Folate-rich diets are highly recommended during pregnancy as these effectively reduce the risk of neural tube defects in newborns [77]. Insufficient folate intake during pregnancy increases the risk of pre-term delivery and fetal growth retardation [78]. Wallock et al. [79] observed a correlation between seminal plasma folate with blood plasma folate; hence, folates are also important for human reproductive health [80].

## 3. Approaches for Improvement of Nutritional Profile

Dietary diversification, food supplements, food fortification, and biofortification are different approaches used for improvement of the nutritional profile of crops to tackle micronutrient deficiency.

### 3.1. Dietary Diversification

Dietary diversification is a food-based strategy that involves consuming a wide range of different foods, especially different plant based foods such as vegetables, fruits, and whole grains. Dietary diversification also uses strategies at the household level, such as preparation of food that involves soaking, fermentation, and germination, as these enhance micronutrient content and bioavailability [81]. Fruits and vegetables rich in promoter substances (ascorbate and β-carotene) that increase mineral absorption should be taken along with a reduced intake of foods rich in anti-nutrients (phytic acid and polyphenols), which inhibit mineral absorption. For example, for iron improvement, foods rich in ascorbic acid (Fe absorption promoter) should be consumed [82,83]. Germination and fermentation can improve iron bioavailability, as these methods increase the activity of phytase enzymes that hydrolyze phytic acid in whole grain cereals and legumes [84]. Levels of folates in diets can be improved by the consumption of naturally folate-rich foods [85] or sprouted seed [86].

### 3.2. Food Supplements

Food supplements are micronutrients consumed in the form of pills, powders, and solutions when diets alone cannot provide an adequate amount of nutrition. Supplementation can be used as a short-term method to improve nutritional health and may be unsustainable for large populations. For example, improvement of folate levels in diets was achieved by the use of folic acid supplements [85,87]. Further, this method had some success with vitamin A and zinc supplementation [88]. Folic acid, iron, and zinc supplements have been helpful for children and pregnant women; however, this method is not cost-effective, especially for low-income consumers [3,89]. Supplementation is a relatively cost-effective method, but may not solve the root cause of micronutrient deficiencies. Supplements for folic acid, zinc, and iron could show different physiological responses and absorption than consuming them in food [3]. Supplementation requires access to medical centers, adequate educational programs, and management of supplies vs. demand, with adequate storage facilities [3,90]. These are manageable in developed countries, but not in rural populations and/or those of developing countries who have little access to these facilities.

### 3.3. Food Fortification

Fortification is the addition of essential micronutrients including vitamins and minerals to foods to improve their nutritional quality. Several food assistance programs by the World Food Program (WFP) are in place using partially pre-cooked and milled cereals and pulses fortified with micronutrients to overcome nutritional deficiencies and provide health benefits with nominal risk. For food fortification with iron, ferrous sulfate, ferrous fumarate, ferric pyrophosphate, and electrolytic iron powder compounds are commonly used [91]. Similarly, food can be fortified with folic acid to improve levels of folates in diets [85,87]. Salt iodization (fortification with iodine) was successfully achieved to reduce the incidence of goiter [92].

### 3.4. Biofortification

Biofortification is a process of improvement of nutritional profile of plant-based foods through agronomic interventions, genetic engineering, and conventional plant breeding (Figure 1).

#### 3.4.1. Agronomic Approaches

Biofortification through agronomic approaches can be achieved by applying mineral fertilizers to the soil, foliar fertilization [93], and soil inoculation with beneficial microorganisms (http://www.fao.org/agriculture/crops/).

##### Mineral Fertilizer

Mineral fertilizers are inorganic substances containing essential minerals and can be applied to the soil to improve the micronutrient status of soil and thus plant quality. The phytoavailability of minerals in the soil is often low; thus, to improve the concentration of minerals in the edible plant tissues, the application of mineral fertilizers with improved solubility and mobility of the minerals is required [93]. This method can be used to fortify plants with mineral elements, but not organic nutrients, such as vitamins, which are synthesized by the plant itself. This method was successfully implemented for Se, I, and Zn, as these elements had good mobility in the soil as well as in the plant [93,94,95,96]. For example, supplementation of inorganic fertilizers with sodium selenate significantly increased Se concentration in various food items, fruits, vegetables, cereals, meat, dairy products, eggs, and fish in Finland [97,98]. Thus, supplementation of fertilizers with sodium selenate proved to be an effective way to increase Se intake in the human population [99]. Similarly, plants were successfully enriched with I and Zn in China and Thailand using inorganic fertilizers, respectively [100]. However, Fe fertilization was not successful due to a low mobility of Fe in soil [101]. The concentration of Zn was increased in field pea grains by either soil application of Zn fertilizer alone or combined with foliar treatments; thus, these methods could be potentially used for the biofortification of field peas [102].

The fertilization strategy for biofortification typically requires regular applications, which could become harmful for environmental health and may limit the availability of other minerals [93,100,103]. Further, soil composition in the specific geographical location, differences in mineral mobility, and the potential of antinutrient compounds limiting mineral bioavailability are also constraints for successful application of this strategy [104,105].

##### Foliar Fertilization

Foliar fertilization is the application of fertilizers directly to the leaves. It could be successful when mineral elements are not available immediately in the soil or not readily translocated to edible tissues [93,106]. Pulse crops were biofortified with micronutrients, Fe, Zn, and Se, through foliar application in various studies that resulted in increased levels of these micronutrients in the harvested grain. Márquez-Quiroz et al. [107] reported increased concentration of Fe (29–32%) in seeds of cowpeas. Ali et al. [108] reported increased Fe concentration (46%) in mungbeans upon foliar application of Fe. Similarly, foliar application of Fe and Zn significantly increased the concentration of these minerals along with protein in seeds of cowpeas [109] and chickpeas [110].

Shivay et al. [111] observed a correlation between Zn uptake and the grain yield of chickpeas following foliar application of Zn, and reported that this approach was better than soil application. Similarly, Hidoto et al. [112] evaluated the effects of three Zn fertilization strategies on five varieties of chickpeas and observed that foliar application was an effective method for Zn biofortification with a greater accumulation of Zn in grain compared to soil application and seed priming. Foliar application of Zn fertilizer for Zn biofortification was also reported in common beans [113,114,115] and field peas [102].

Increased concentration of Se was reported in seeds of peas [116], chickpeas [117], common beans [118], and lentils [119] upon foliar application of Se fertilizers. Further, increased concentration of I was observed in various crops by foliar application, and this could prevent I deficiency in human populations with low dietary I intake [55].

##### Plant Growth Promoting Microorganisms

Rhizobia, mycorrhizal fungi, actinomycetes, and diazotrophic bacteria are beneficial soil microorganisms associated with plant roots by symbiotic association, and these protect plants by various methods such as promotion of nutrient mineralization and availability and production of plant growth hormones [120]. Though these are naturally present in the soil, their populations can be enhanced by inoculation or agricultural management practices. Various plant growth-promoting (PGP) soil microorganisms including *Enterobacter*, *Bacillus*, and *Pseudomonas* can be exploited to increase the phytoavailability of micronutrients. These are used mostly as seed inoculants and enhance plant growth through the production of growth hormones, antibiotics, chitinases, and siderophores and the induction of systemic resistance and mineralization [121]. PGP microorganisms chelate iron via the production of siderophore compounds, solubilize phosphorus, and inhibit growth of pathogens [122,123], thus playing a significant role in soil fertility and iron fortification. PGP microbes are usually present in soil, compost, and decomposing organic materials and provide an economical and harmless method for increasing crop production and improve environmental and soil health [124].

Numerous studies have shown increased concentrations of Fe, Se, and Zn using microorganism inoculants via mycorrhizal associations [125,126,127]. Further, enhancement of nitrogen fixation, plant growth, and grain yield have been reported in legumes including chickpeas, soybeans and peas by colonization of *Pseudomonas* sp., *Brevibacterium* sp., *Bacillus* sp., *Enterobacter* sp., and *Acinetobacter* sp. in their roots and nodules [128,129,130,131,132].

In chickpeas, inoculation of PGP actinobacteria increased the concentration of seed minerals including Fe (10–38%) and Zn (13–30%) compared to control (uninoculated) plants [133]. Similarly, arbuscular mycorrhizal fungi field inoculation improved the nutritional profile of chickpea grains by increasing Fe and Zn concentration along with yield and protein content [134]. Khalid et al. [135] reported that application of PGP rhizobacteria with Fe compound (FeSO_4_) in soil increased iron concentration (up to 81%) in chickpeas compared to a control, and suggested a potential role of microorganisms in the additional uptake of Fe from soil upon supplementation with Fe. Gopalakrishnan et al. [124] inoculated the seeds before sowing and the field plots of chickpeas and pigeonpeas every 15 days until the flowering stage with seven strains of bacteria, *P. plecoglossicida* (SRI-156), *B. antiquum* (SRI-158), *B. altitudinis* (SRI-178), *E. ludwigii* (SRI-211), *E. ludwigii* (SRI-229), *A. tandoii* (SRI-305), and *P. monteilii* (SRI-360), and observed that these bacterial strains significantly improved several growth parameters including nodule, pod number, and grain yield compared to un-inoculated control plots. The harvested grains showed increased concentration of minerals including Fe and Zn. Iron concentration was increased up to 18 and 12%, whereas the concentration of Zn was increased up to 23 and 5% in chickpeas and pigeonpeas, respectively.

#### 3.4.2. Genetic Engineering

Biofortification through genetic engineering is an alternative approach when variation in the desired traits is not available naturally in the available germplasm, a specific micronutrient does not naturally exist in crops, and/or modifications cannot be achieved by conventional breeding [136,137]. This approach was supported by the availability of fully sequenced genomes in various crops in recent years. Along with increasing the concentration of micronutrients, this approach can also be targeted simultaneously for removal of antinutrients or inclusion of promoters that can enhance the bioavailability of micronutrients [93,106,138]. This approach had not only utilized genes associated with various metabolic pathways operated in plants, but also from bacteria and other organisms [139,140]. Development of transgenic crops requires a substantial investment during the initial stage, but this could be a sustainable approach that has the potential to target large populations, especially in developing countries [96,103,141].

Several crops have been successfully modified using a transgenic approach to overcome a micronutrient deficiency. For example, enhanced accumulation (3 to 4 times) of Fe was noted in rice via expression of the iron-storage protein, ferritin [142,143]. In rice, the national levels for Fe and Zn biofortification nutrition targets were attained under field settings in the Philippines and Colombia [144]. The genetically engineered rice (golden rice) was developed to produce β-carotene (pro-vitamin A) to fight against vitamin A deficiency [145]. Recently, transgenic multivitamin corn was produced by the simultaneous modification of three distinct metabolic pathways to increase the levels of three vitamins, i.e., β-carotene (169-fold), ascorbate (6-fold), and folate (2-fold), in the endosperm, and this could pave the way to develop nutritionally complete cereals [146]. Using metabolic engineering, the folate concentration was increased in tomato and rice [85,147]. Storozhenko et al. [148] reported more than 100-fold increase in folate concentration in rice by overexpression of *Arabidopsis thaliana* pterin and para-aminobenzoate genes, precursors of the folate biosynthesis pathway, whereas Hossain et al. [149] reported a two- to four-fold increase in *Arabidopsis* by overexpression of the gene involved in pterin biosynthesis.

To the best of our knowledge, there are no examples of biofortified pulse crops developed through a transgenic approach for Fe, Zn, Se, I, carotenoids, or folates in the available literature. However, a genetic engineering approach has been applied in pulse crops for improvement of other nutritional profile. For example, the concentration of the essential amino acid methionine was significantly increased in transgenic common bean plants (up to 23%) by expression of a methionine-rich storage albumin from the Brazil nut [150], and in the concentration of methionine in transgenic lupins (up to 94%) by expressing a sunflower seed albumin gene [151].

In recent years, targeted gene editing technologies using artificial nucleases, zinc finger nucleases (ZFNs), transcription activator-like effector nucleases (TALENs), and the clustered regularly interspaced short palindromic repeat (CRISPR)/CRISPR-associated protein 9 (Cas9) system (CRISPR/Cas9) have given rise to the possibility to precisely modify genes of interest, and thus have potential application for crop improvement [152,153]. These technologies have been used in various crops including rice [154,155], wheat [156], and tomatoes [157]. Recently, CRISPR/Cas9 and TALENs technologies were used to generate mutant lines for genes involved in small RNA processing of *Glycine max* and *Medicago truncatula* [158]. Similarly, CRISPR/Cas9-mediated genome editing technology was used in cowpeas to successfully disrupt symbiotic nitrogen fixation (SNF) gene activation [159]. These findings pave the way for applicability of use of gene editing technologies for various traits of interest in legumes.

#### 3.4.3. Plant Breeding

Limitations in the long-term effectiveness and sustainability of fertilizer approaches necessitates the development of economical and longstanding strategies for increasing micronutrient density in plants. Genetic engineering technology to produce genetically modified plants with desirable traits has been used in corn, rice, wheat, and soybeans. This can be an effective approach for crop improvement; however, political opposition to GMOs in many countries, a complex legal framework for the acceptance and commercialization of transgenic crops, along with expensive and time-consuming regulatory processes are the major limitations of this method [100,160,161]. For example, golden rice has been available since the early 2000s and has the potential to deliver more than 50% of the estimated average requirement for vitamin A, but unfortunately it has not been commercially introduced in any country to date due to risk factors involved in the regulatory approval processes [162,163]. In developed and developing countries, several groups endorse the arguments made by Greenpeace that approval of golden rice will allow multinational corporations to control developing countries’ food supplies and pose risks to human health and the environment (https://gmo.geneticliteracyproject.org/).

Restrictions on the use of genetically modified crops in many countries prompted HarvestPlus to take the initiative to address micronutrient deficiencies through conventional plant breeding [20,22]. Biofortification through plant breeding is a cost-effective and sustainable approach that can improve the health status of low-income people globally [19,21,147]. This approach has been used to control deficiencies of micronutrients including carotenoids, Fe, and Zn [96,164].

HarvestPlus was started in 2003 by the initiative of the Consultative Group on International Agricultural Research (CGIAR) to target several major food crops, including rice, common beans, cassava, maize, sweet potatoes, pearl millet, and wheat in Asia and Africa, to enrich them with three major nutrients, Fe, Zn, and vitamin A through an interdisciplinary and global alliance of scientific institutions and implementing agencies [165,166]. HarvestPlus is part of the CGIAR Research Program on Agriculture for Nutrition and Health (A4NH). The HarvestPlus program is administered by joint venture of the International Center for Tropical Agriculture and the International Food Policy Research Institute and provides global leadership on biofortification evidence and technology. The UK Government, the Bill and Melinda Gates Foundation, the US Government’s Feed the Future initiative, the EU Commission, and donors to A4NH are principal investors [163] https://www.harvestplus.org.

Conventional plant breeding approaches can benefit not only large populations but also people living in relatively remote rural areas who have limited access to commercially marketed fortified foods [19,21,22,167]. This approach requires a one-time investment in plant breeding and can be grown and multiplied across years by farmers at virtually zero marginal cost. Recurrent costs are low, and germplasms can be available internationally without any adverse effect on productivity and health; thus, biofortification has widespread public acceptance [20,21,100,168].

Genetic diversity is required in the gene pool to achieve success in biofortification through the plant breeding approach. Several studies have shown substantial variation in the concentration of minerals and vitamins in various crops [93,106]. Parental genotypes with high micronutrient concentration can be identified by screening a wide range of germplasms, and these can be utilized in making crosses, genetic studies, and the development of molecular markers to facilitate marker-assisted selection in breeding. Promising lines can be tested at multiple locations to determine the genotype X environment interaction (G X E) [163]. These can be submitted to national government agencies for testing for agronomic performance and release after robust regional testing across multiple locations over multiple seasons [163]. Using the above-mentioned strategy, various studies in pulse crops have identified substantial variation in the available gene pools and recombinant inbred line populations developed from promising parental genotypes. These were utilized for identifying genomic regions and associated markers for their use in marker-assisted selection. Recent research advances in pulse crops for improvement of key micronutrients, including Fe, Zn, Se, I, carotenoids, and folates, through conventional plant breeding approaches, will be discussed in the following section.

## 4. Recent Research Advances for Biofortification of Pulse Crops

### 4.1. Iron

In several studies, a wide range of variation in Fe concentration has been observed in peas, chickpeas, common beans, mungbeans, and lentils (Table 1). Significant genetic variability in Fe concentration was observed in the core collection of common beans [169,170], common bean populations [171,172,173], and a large collection of lentil accessions [174,175]. Cultivars of lentils (18), peas (17), common beans (10), and chickpeas (8) grown at several locations in Southern Saskatchewan (2005–2006) had Fe concentration of 75.6–100 mg kg^−1^, 47.7–58.1 mg kg^−1^, 57.7–80.7 mg kg^−1^, and 48.6–55.6 mg kg^−1^, respectively [11]. A 100 g serving of any of these pulse crops provided over 50% of the RDA for Fe. A substantial variation in concentration of Fe was observed in 94 diverse accessions each of chickpeas [12] and peas [13]. Further, Diapari et al. [12] identified several chickpea accessions with high Fe concentration (52–60 mg kg^−1^) that could be utilized for the development of cultivars with high Fe concentration. Significant variation was observed in Fe concentration (35–87 mg kg^−1^) in mungbean lines commonly grown in South Asia [176]. In a recent study, Dissanayaka [29] reported significant variation in a pea genome wide association study (GWAS) panel of 177 accessions evaluated at Saskatoon and Rosthern Saskatchewan, Canada, and Fargo, North Dakota, USA.

The environment significantly affected the concentration of minerals in field peas, chickpeas, common beans, and lentils grown at different locations in Saskatchewan, Canada [11]. Diapari et al. [13] and Dissanayaka [29] reported a significant effect of genotypes, year, and location with higher Fe concentration at Rosthern compared with Saskatoon in peas.

Effect of genotypes was also significant in lentils [25,28] and chickpea [12,28]. Along with environment, Fe concentration was also affected by varieties and locations. For example, Ariza-Nieto et al. [177] in common beans and DellaValle et al. [178] in lentils observed differences in Fe concentration at the same location due to varieties, whereas Moraghan et al. [179] reported higher Fe concentration in seeds harvested from acid soil compared to calcareous soil. Diapari et al. [13] reported that the genetic factor was responsible for substantial variation in Fe concentration in pea seeds grown at different locations in Saskatchewan.

Several SNP markers associated with Fe concentration were identified in peas [13,26,27,29], chickpeas [12,23], and lentils [24,25] that can be used in marker-assisted selection (Table 1). In peas, Diapari et al. [13] reported an association of nine SNPs with Fe concentration in 94 diverse accessions, whereas Dissanayaka [29] reported significant association of three SNP markers with Fe concentration in a pea GWAS panel of 177 accessions. Ma et al. [26] identified five QTLs for Fe concentration in a pea population developed from “Aragorn” (PI 648006) and “Kiflica” (PI 357292). Similarly, Gali et al. [27] observed several QTLs for seed iron concentration on four LGs of pea population PR-02 (Orb X CDC Striker) and six LGs of PR-07 (Carerra X CDC Striker).

In chickpeas, Diapari et al. [12] identified two SNPs on Chromosome (chr) 4 and one each on chr1 and chr6 in 94 diverse accessions, whereas Upadhyaya et al. [23] observed associations of six QTLs for Fe concentration on Chromosomes 1, 3, 4, 5, and 7 in a population developed from a cross between ICC 4958 and ICC 8261. In lentils, Aldemir et al. [24] identified 21 QTLs for Fe concentration on several linkage groups (three each on LG1, LGII, and LGVII, six QTLs on LGIV, four QTLs on LGV, and two QTLs on LGVI), whereas Khazaei et al. [25] reported nine QTLs in a panel of 138 accessions, and two of them were tightly linked to Fe concentration in Chromosomes 5 and 6. Similarly, several QTLs were identified for Fe concentration in common bean populations, DOR364 X G19833 [171], G14519 X G4825 [172], and G21242 X G21078 [173].

### 4.2. Zinc

Like Fe, a wide range of variation in Zn concentration was observed in peas, chickpeas, common beans, and lentils (Table 1). A significant variation in Zn concentration was noted in 94 diverse chickpea accessions evaluated under field conditions in Saskatchewan, Canada [12]. This study identified three kabuli type accessions, CDC Verano, ILC 2555, and FLIP85-1C (43–48 mg kg^−1^), and two desi type accessions, FLIP97-677C and FLIP84-48C (42 and 41 mg kg^−1^), with the greatest Zn concentrations.

A substantial variation in Zn concentration was observed in common beans (24.8–33.3 mg kg^−1^), peas (27.4–34 mg kg^−1^), chickpeas (21.1–28.3 mg kg^−1^), and lentils (36.7–50.6 mg kg^−1^) grown in two years (2005–2006) at various locations in Saskatchewan [11], and each of these provided over 50% of the RDA in 100 g of dry pulses. Significant variation for Zn was observed in the core collection of common beans (>2400) [169,170], three common bean populations [171,172,173], lentil accessions (>1600) [174,175], 20 mungbean lines [176], a panel of 94 pea accessions [13], a panel of 177 pea accessions [29], and two pea populations, PR-02 and PR-07 [27].

In various studies, the effect of genotypes, year, and/or location was significant for Zn concentration in peas [13,29], chickpeas [12,28], and lentils [25,28]. Further, Zn concentration was often positively correlated with Fe concentration [12,13,25,26,29].

Several QTLs and/or SNP markers were identified for Zn concentration in common beans, chickpeas, lentils, and peas (Table 1). For example, QTLs were identified for Zn concentration in three common bean populations [171,172,173]. Five SNPs were identified for Zn concentration in chickpeas, and these were located on Chromosomes 1, 4, and 7 [12]. Further, in chickpeas, Upadhyaya et al. [23] observed an association of eight genomic loci for Zn concentration. In lentils, Khazaei et al. [25] reported twelve SNP markers for Zn concentration in a panel of 138 accessions grown at two locations in Saskatchewan, Canada in 2013–2014.

Using a GWAS, Diapari et al. [13] found two SNPs for Zn on LG III in 94 pea accessions. Ma et al. [26] identified five QTLs for Zn concentration on LGs II, III, V, and VII in a pea population developed from Aragorn X Kiflica. Similarly, Gali et al. [27] identified one QTL each on LG1a and LG3b, two QTLs on LG6 in PR-02, and numerous QTLs on various LGs (1a, 1b, 2b, 3b, 4, and 7a) in the PR-07 pea population. In a recent study, Dissanayaka [29] reported a significant association of seven SNP markers with Zn concentration in a pea GWAS panel of 177 accessions. Further, the SNP marker Sc1512_36017 was co-localized with Sc11336_48840 on LG IIIb in PR-07. In a previous study, Sc11336_48840 was identified as the flanking marker of a QTL for seed Zn concentration [27].

### 4.3. Selenium

Soil and weather conditions play important roles in Se concentration in harvested pulse crop seeds. Among soil factors, aeration, water availability, pH, and texture are important, as these affect the availability of Se [195]. Diapari et al. [13] observed that variation in Se concentration in peas was mainly due to environment, whereas the effect of genotype was minimal (only 2.7% of the total). Se concentration was higher at the Saskatoon location than Rosthern, and differences in Se concentration were not significant across the 94 genotypes.

Soils of Saskatchewan are generally rich in Se, so pulses grown in this region provide a natural dietary source of this element [180,182,196]. Lentils grown in the Dark Brown and Brown soil zones of Western Canada had a high concentration of Se (425–672 μg kg^−1^) [180] (Table 1). In comparison, lentils grown in six major lentil-producing countries, Nepal (180 μg kg^−1^), Southern Australia (148 μg kg^−1^), Turkey (47 μg kg^−1^), Morocco (28 μg kg^−1^), Northwestern USA (26 μg kg^−1^), and Syria (22 μg kg^−1^), had a substantially lower Se concentration [181]. A wide range of variation was observed for Se concentration in common beans (381–500 μg kg^−1^), peas (405–554 μg kg^−1^), chickpeas (629–864 μg kg^−1^), and lentils (990–1637 μg kg^−1^) grown at several locations in Saskatchewan [11]. A 100 g dry weight of any of these pulses could provide 100% of the RDA. Nair et al. [176] observed significant variation for Se concentration (210–910 µg kg^−1^) in mungbean lines grown in two environments near Hyderabad, India.

Total Se concentration varied from 373 to 519 µg kg^−1^ in 17 field pea cultivars grown at six locations for 2 years in Saskatchewan [182], and this provided 68–94% of the RDA upon the serving of 100 g peas. Evaluation of 80 pea breeding lines obtained from Australia, Czech Republic, Serbia, and the United States had relatively low concentration of Se; however, these lines showed greater concentration of Se when planted in Saskatoon [182]. Similarly, Gali et al. [27] observed a considerable variation in Se concentration in pea populations, PR-02 and PR-07, with a greater range in variation among RILs at the Saskatoon location compared with Rosthern. Compared to Fe and Zn, Dissanayaka [29] observed a substantially different pattern for Se at different locations, with a high coefficient of variation, and the concentration varied from 0.06 to 8.75 ppm. The effects of genotype, genotype × year, and genotype × location were not significant, with the exception of the genotype effect at the trial grown in 2014 at Fargo, North Dakota.

Several QTLs were identified for Se concentration in the PR-02 pea population on LGs 4a, 5a and 7, and in the PR-07 pea population on LG4 and 5b [27] (Table 1). Using a pea GWAS panel of 177 accessions, 44 significant SNP markers were identified for Se concentration, but the majority of the markers were not common among the location-years [29], and this could be due to substantial variation in Se concentration and high coefficient of variation at different locations.

### 4.4. Iodine

Several studies have reported various methods such as foliar fertilization and application of salt in soil and/or irrigation water for biofortification of crops with iodine; however, little information is available on within-species variation. The consumption of cereal-based foods with low I concentration is the major cause of I deficiency in humans [55,56,197]. Compared to grain, biofortification of leaves and leafy vegetables could be easily achieved due to translocation of the majority of I to xylem tissues, thus the majority of research is focused on I biofortification of vegetables instead of grains [56,198,199,200].

### 4.5. Carotenoids

Several studies have reported a carotenoid profile in pulse crops [183,185,186,201,202,203,204] (Table 1). Lutein, zeaxanthin, and β-cryptoxanthin were reported in chickpeas [183,202], whereas violaxanthin, lutein, and β-carotene were reported in field peas [201,203]. Thavarajah and Thavarajah [184] reported a high concentration of carotenoids, beta-carotene (166–431 μg/100 g), canthoxanthine (21–68 mg/100 g), and xanthophyll (9–20 mg/100 g) in 10 chickpea genotypes grown in Minot, North Dakota.

Ashokkumar et al. [185] evaluated carotenoids profile in 12 pea and 8 chickpea cultivars grown at multiple locations in Saskatchewan, Canada, using high performance liquid chromatography with a diode array detector. This method is sensitive, reliable, and accurate for the separation and quantification of putative carotenoids. In peas, the concentration of carotenoids was greatest in cotyledon, followed by embryo axis and seed coat. Green cotyledon cultivars (16–21 µg g^−1^) had generally higher concentrations compared to yellow cotyledon cultivars (7–12 µg g^−1^). Lutein was the major component (11.45 µg g^−1^) followed by violaxanthin (0.52 µg g^−1^), β-carotene (0.47 µg g^−1^), and zeaxanthin (0.16 µg g^−1^). In kabuli type chickpea cultivars, carotenoid concentration was highest in the cotyledon, followed by the embryo axis and seed coat, whereas in the desi type, the seed coat, followed by the cotyledon and embryo axis, had the highest carotenoid concentration. Lutein (7.70 μg g^−1^) was the major component followed by zeaxanthin (5.76 μg g^−1^), β-carotene (0.40 μg g^−1^), and violaxanthin (0.05 μg g^−1^).

In subsequent work, Ashokkumar et al. [186] observed a wide range of variation in concentration of carotenoids in genetically diverse pea (94) and chickpea (121) accessions grown at multiple locations in Saskatchewan, Canada. In the peas, the concentration of lutein was highest (11.2 µg g^−1^) followed by β-carotene (0.5 µg g^−1^), zeaxanthin (0.3 µg g^−1^), and violaxanthin (0.3 µg g^−1^), whereas in the chickpeas, the concentration of lutein (8.2 µg g^−1^) was highest followed by zeaxanthin (6.2 µg g^−1^), b-carotene (0.5 µg g^−1^), β -cryptoxanthin (0.1 µg g^−1^), and violaxanthin (0.1 µg g^−1^). Green cotyledon peas and desi chickpeas had a greater carotenoid concentration than yellow cotyledon peas and kabuli chickpeas, respectively. Pea and chickpea accessions with high carotenoid concentration that can be utilized in future breeding were identified. In five chickpea cultivars with different cotyledon colors, total carotenoid concentration varied from 22 μg g^−1^ (yellow cotyledon kabuli) to 44 μg g^−1^ (green cotyledon desi), with lutein and zeaxanthin as major components [187]. In a recent study, a wide range of total carotenoid concentration was observed in three F_2_ populations developed by crossing cultivars with different cotyledon and seed coat colors, CDC Jade X CDC Frontier (14.9–58.1 µg g^−1^), CDC Cory X CDC Jade (1.9–77.6 µg g^−1^), and ICC4475 X CDC Jade (21.6–83.7 µg g^−1^) [188].

Total carotenoids (5.8–26.9 μg g^−1^) and β-carotene (2.6 μg g^−1^) were greater in peas [186] compared with potato accessions (1.4–14.3 μg g^−1^) [205] and golden rice endosperm (1.6 μg g^−1^) [206]. Similarly, in 121 chickpea accessions, the concentration of total carotenoids (15.0 μg g^−1^) was three times greater than in 42 banana accessions (4.7 μg g^−1^) [207] and 37 potato accessions (4.4 μg g^−1^) [208]. Of all carotenoids identified, lutein was the major compound in chickpeas [183,185,186,202] and peas [185,186,201,203,204]. Further, lutein has positive correlations with chlorophyll concentration in peas [203] and zeaxanthin concentration in chickpeas [183].

Previously, four QTLs for beta-carotene concentration and a single QTL for lutein concentration were detected in chickpeas [183] (Table 1). Using a GWAS, Rezaei et al. [187] identified 32 candidate genes involved in isoprenoid and carotenoid pathways across all eight chromosomes of chickpeas. They observed positive correlation between the expression of genes of carotenoid biosynthesis with various carotenoid components. In a subsequent study, several QTLs were identified for total carotenoids and individual components in three F_2_ populations, CDC Jade × CDC Frontier (8 QTLs on LGs 1, 5, and 8), CDC Cory × CDC Jade (5 QTLs on LG 8), and ICC4475 × CDC Jade (5 QTLs on LGs 3 and 8) [188]. Further, several candidate genes associated with carotenoid components were observed with a major gene for cotyledon color on LG 8 in each population.

### 4.6. Folates

A wide exploration of available genetic resources is necessary to identify the rich source of folates for their potential use for the biofortification of pulse crops. Various methods have been employed to quantify folates from different food sources including pulse crops, and these methods include microbiological assays [209,210,211], liquid chromatography (LC) coupled with fluorescence detection (FD) [87,192], and mass spectrometry (MS) detection [15,30,191,193,194,212,213,214].

Previously, researchers have identified a wide range of variation in folates quantified from pulse crops (Table 1). For example, using a microbiological assay, Han and Tyler [190] reported 24.9–64.8 µg/100 g folates in green cotyledon peas and 23.7–55.6 µg/100 g in yellow cotyledon peas grown at multiple locations in Saskatchewan, Canada. Vahteristo et al. [189] and Hefni et al. [87] observed the folate concentration of 59 and 52 µg/100 g using LC-MS in vegetable peas consumed in Finland and dry green peas consumed in Egypt, respectively. Rychlik et al. [191] reported variable folate concentration in different pulse crops—275 µg/100 g (chickpeas), 106–164 µg/100 g (white beans), 110–154 µg/100 g (green lentils), and 10–20 µg/100 g (peas) using an LC-MS method, whereas Sen Gupta et al. [192] reported a higher total folate concentration in field peas (41–202 µg/100 g) grown in the USA compared to chickpeas (42–125 µg/100 g) using LC-FD.

Using ultra-performance LC (UPLC), six folate monoglutamates were quantified in four pulse crops, and the total folate concentration was the highest in chickpeas (351–589 µg/100 g), followed by common beans (165–232 µg/100 g), lentils (136–182 µg/100 g), and pea (23–30 µg/100 g) [15]. This method provided high accuracy in quantification of specific folates with the use of isotopically labeled internal standards. Zhang et al. [193] identified eight folate monoglutamates using an optimized one-step extraction approach in peas, chickpeas, beans, and lentils using UPLC-MS. They also observed the highest folate concentration in chickpeas and the lowest concentration in peas. Zhang et al. [194] quantified eight folate monoglutamates in six wild lentil species and one cultivated species using UPLC-MS. They observed that wild lentil species (195–497 μg/100 g) had generally higher folate concentration than cultivated genotypes (174–361 μg/100 g). Most recently, Jha et al. [30] quantified five folate monoglutamates in 85 diverse pea accessions originating from worldwide sources using UPLC-MS, and the results indicated a wide range of variation in the concentration of the sum of folates (14–55 µg/100 g dry seed weight).

Tetrahydrofolate (THF), 5-methyltetrahydrofolate (5-MTHF), and 5-formyltetrahydrofolate (5-FTHF) were the most abundant in common beans, lentils, chickpeas, and peas [15,30,208]. Further, 5-MTHF represented 56% of the total folate concentration in peas [15,30], whereas 5-MTHF and 5-FTHF represented 35–39% and 33–51% of the total folates in common beans, lentils, and chickpeas, respectively [15]. Other studies also reported 5-MTHF as the predominant form of folate in common beans [87,215], lentils [87,191], and chickpeas [87]. 5-MTHF was also the major folate in cereals, vegetables, fruit, bread, milk, and meat products [87,216], and in humans [217]. Scaglione and Panzavolta [218] reported various advantages to using naturally occurring 5-MTHF over synthetic folic acid. For example, it helps in the prevention of the potential negative effects of unconverted folic acid in peripheral circulation. Thus, 5-MTHF may be the most important folate that could be targeted for improvement by breeders. Jha et al. [30] identified several pea accessions with greater folate concentration including MPG87 (42 µg/100 g), Kahuna-NIAB (40 µg/100 g), and OZP0902 (50 µg/100 g).

Using a GWAS, five SNP markers were associated with the sum of folates, and fifteen, eight, and three SNP markers were associated with major individual folates 5-MTHF, 5-FTHF, and THF, respectively [30] (Table 1). Further, SNP markers Sc_6992_86348 and Sc_3060_11265 were validated in an additional 24 accessions, and these markers have potential for marker-assisted selection in pea breeding.

## 5. Status of Biofortification

In recent years, several crops with increased micronutrient concentration have been introduced in several developing countries, and this has helped in overcoming nutrient deficiency in the target population. For example, the introduction of the orange sweet potato biofortified with β-carotene increased vitamin A intake among children and women in Mozambique [219] and Uganda [220], and maize biofortified with provitamin A increased the concentration of vitamin A in 5–7-year-old children in Zambia who consumed it for three months [221]. Similarly, serum ferritin and total body iron were improved in iron-deficient adolescent boys and girls from Maharashtra, India, who consumed Fe-biofortified pearl millet flat bread for four months [222]. Regarding pulse crops, the consumption of Fe biofortified beans for 4.5 months improved the hemoglobin and total body iron in iron-depleted university women in Rwanda [223].

By the end of 2016, more than 150 biofortified varieties of 10 crops had been released in 30 countries, and these are consumed by more than 20 million people in developing countries [163]. To date, HarvestPlus has released or tested more than 290 varieties of 12 staple food crops including vitamin A orange sweet potato, iron beans, iron pearl millet, vitamin A yellow cassava, vitamin A orange maize, zinc rice, and zinc wheat in 60 countries (www.harvestplus.org). Iron beans are delivered in Rwanda and Democratic Republic of Congo, zinc rice is in Bangladesh, and zinc wheat is in India and Pakistan.

Among pulse crops, HarvestPlus has released 10 Fe-biofortified bean varieties each in Rwanda (RWR 2245, RWR 2154, MAC 42, MAC 44, CAB 2, RWV 1129, RWV 3006, RWV 3316, RWV 3317, and RWV 2887) and the Democratic Republic of Congo (COD MLB 001, COD MLB 032, HM 21-7, RWR 2245, PVA 1438, COD MLV 059, VCB 81013, Nain de Kyondo, Cuarentino, and Namulenga) (www.harvestplus.org). Similarly, several varieties of lentils with high iron and zinc have been released by HarvestPlus and The International Center for Agricultural Research in the Dry Areas (ICARDA) in various countries: seven in Nepal (ILL 7723, Khajurah-1, Khajurah-2, Shital, Sisir Shekhar, Simal), five in Bangladesh (Barimasur-4, Barimasur-5, Barimasur-6, Barimasur-7, and Barimasur-8), two each in India (L4704, Pusa Vaibhav) and Syria (Idlib-2, Idlib-3), and one in Ethiopia (Alemaya). For the effective delivery and production of these crops, HarvestPlus works closely with various public and private organizations [163]. For example, in Rwanda, HarvestPlus with the help of Rwanda Agriculture Board (RAB) facilitated the production of bean seeds through contracted farmers and cooperatives and acquired about 80% of certified seeds during 2011–2015.

Biofortification to enrich nutrient profile of pulse crops is one of the major goals in the pulse crop breeding program at the Crop Development Centre (CDC), University of Saskatchewan, which was established in 1971 with the objectives to improve existing crops and develop new crops (https://agbio.usask.ca/research/centres-and-facilities/crop-development-centre.php#MoreAbouttheCDC). In recent years, several projects have been undertaken to evaluate pulse crops for the profiling of folates, carotenoids, polyphenols, Fe, Zn, and Se.

## 6. Challenges and Future Strategies for Biofortification

A greater micronutrient density and a high yield are prerequisites for effective biofortification, and these crops must be adopted by farmers and consumed by the target population [21]. Bouis and Saltzman [163] outlined three important challenges for HarvestPlus to reach one billion people by 2030, i.e., building consumer demand, mainstreaming biofortified traits into public and private breeding programs, and integrating biofortification into public and private policies.

Various factors such as genetic diversity in the gene pool, the reduction of antinutrients (especially phytate and polyphenols), and increasing the concentration of promoter substances including certain amino acids (cysteine, lysine, and methionine) and ascorbic acid (vitamin C), which enhance the absorption of essential minerals, and/or high yield, are key for the success of biofortification strategies [93,167].

Narrow genetic variation in the plant gene pool, a long-development time for generating cultivars with a desired trait, and the dependence on the phytoavailability of the mineral nutrients in the soil are limitations for conventional breeding approach [138].

The issue of narrow genetic variation for micronutrient concentration might be overcome by the use of wild germplasm and land races, which may contain a high variation in micronutrient concentration [93,106,138].

For efficient biofortification, the focus should be on increasing the bioavailability of micronutrients simultaneously with increase in their concentration. This can be achieved by increasing the concentration of promoters that stimulate the absorption of minerals and by reducing the concentrations of antinutrients that interfere with absorption [93].

Vitamin E, vitamin D, vitamin C, choline, niacine, and provitamin A are considered promoter substances and stimulate the absorption of Se, Ca P, Fe, Zn, methionine, and tryptophan [224]. In contrast, certain antinutrients including phytate and certain polyphenols reduce the bioavailability of micronutrients in crops [93]. Phytate, a form of phosphorus stored in seed, is not digested by humans or monogastric animals [225]. During digestion, it can bind to iron and zinc and thus restrict their absorption [226]. The concentration of phytate can be controlled by identifying low phytate lines by germplasm screening [227], manipulating the biosynthesis of phytate via mutation of a myo-inositol kinase (MIK) gene [228], and overexpressing phytase, a phytate degrading enzyme [229].

In the recent past, low-phytate lines in pulse crops have been developed and characterized to reduce the concentration of phytate and thus improve mineral absorption [225,230,231,232,233]. Warkentin et al. [225] developed low-phytate pea lines, 1-150-81 and 1-2347-144, using chemical mutagenesis of cultivar CDC Bronco, a high-performing pea variety. They observed an approximately 60% reduction in phytate phosphorus in low-phytate lines with an increase in inorganic phosphorus. However, these lines had a slightly lower seed weight and a lower yield compared to CDC Bronco. Nevertheless, these lines are being used to breed for improvement of phosphorus and micronutrient bioavailability, along with high grain yield. Subsequently, Liu et al. [226] evaluated the effects of phytate and seed coat polyphenols on the bioavailability of iron using low-phytate pea lines (1-150-81 and 1-2347-144). The iron bioavailability (FEBIO) was 1.4–1.9 times greater in low-phytate lines compared to normal phytate varieties. Further, pigmented seed coat pea showed a seven times lower FEBIO than non-pigmented seed coats; however, the removal of seeds coats increased the FEBIO up to six times. To understand the genetic basis of the low phytic acid (*lpa*) mutation in the pea, Shunmugam et al. [232] amplified a 1530 bp open reading frame of *myo*-inositol phosphate synthase (MIPS), the rate-limiting step in the phytic acid biosynthesis pathway, from CDC Bronco and two *lpa* pea genotypes, 1-150-81 and 1-2347-144. They did not observe any difference in coding sequence in *MIPS* between CDC Bronco and *lpa* genotypes and noticed that mutation in *MIPS* did not cause the *lpa* trait in pea lines.

Various studies in common beans suggested that *lpa* lines can improve iron bioavailability by reducing the phytic acid level up to 90% [230,234]. Homozygous *lpa* mutant line (*lpa-280-10*) was isolated in common beans from a mutagenized population, and this mutant had 90% less phytic acid and higher free Fe in the seeds compared to the wild type [230]. Further, at the molecular level, it was observed that a recessive mutation was responsible for the *lpa* character. Panzeri et al. [231] mapped the *lpa1*(280-10) mutation and identified and sequenced a candidate gene in common beans for comparison with the soybean genome. They observed that the *lpa1*(280-10) mutation co-segregated with the mutated multidrug resistance-associated protein (MRP) type ATP-binding cassette transporter gene *(Pvmrp1)*, which is orthologous to the *lpa* genes of *Arabidopsis AtMRP5* and maize *ZmMRP4*. They further observed that a defective *Mrp1* gene caused an *lpa1* mutation in common beans that downregulates the phytic acid pathway at the transcriptional level and thus reduced seed myo-inositol. Recently, a new *lpa* line influencing the *PvMRP1* phytic acid transporter was identified in common beans using ethyl methane sulfonate mutagenesis [233]. Further, *PvMRP* promoters were characterized in *Arabidopsis thaliana* and *Medicago truncatula* transgenic plants.

Polyphenols are secondary metabolites including flavonoids and proanthocyanidins [235] and provide protection against various fungal pathogens [236]. They are natural sources of antioxidants in the human diet and are present in fruits, vegetables, cereals, and legumes [237,238]. Previously, all polyphenols were considered as inhibitors of Fe bioavailability in humans. A recent study by Hart et al. [239] reported that four polyphenols inhibited Fe uptake, whereas four other polyphenols promoted Fe uptake upon evaluating the effect of polyphenols present in black bean seed coats on Fe uptake using Caco-2 cells (human cell line). They further concluded that specific polyphenols (promoter of Fe uptake) can be targeted in future breeding for improved Fe bioavailability. Jha et al. [240] detected 30 polyphenols in a recombinant inbred line population developed from crossing pea cultivars CDC Amarillo (white flower) and CDC Dakota (purple flower). Among 30 polyphenols, catechin, 3,4-dihydroxybenzoic acid, and kaempferol 3-glucoside were present in all pea lines, and these were considered promoters of Fe uptake by Hart et al. [239]. Thus, promising accessions having Fe promoter polyphenols can be identified via a wide exploration of germplasms for developing cultivars with additional health benefits.

Postharvest processing can also play an important role in efficient utilization of biofortified crops, as a substantial amount of minerals from the diet can be lost by milling or polishing [241] and cooking. Therefore, efforts should be made to retain the micronutrient concentration in edible seeds, and their absorption by the consumer after processing and cooking [242]. Retention of zinc content after cooking in biofortified rice varieties produced either through traditional breeding or genetic engineering approaches has been discussed in detail by Tsakirpaloglou et al. [243].

Iodization of salt was not enough to overcome I deficiency due to several factors such as the unavailability of iodized salt for all households, the volatilization of I during cooking, and insufficient consumption due to health issues [55,93,200,244]. Hence, for successful biofortification, further research was needed to identify traits that control uptake, mobilization, and retention of I in the plant, and these can be manipulated in plant breeding or using a genetic engineering approach [56].

In pulse crops, growth and productivity are affected by various abiotic and biotic stresses, which can result in significant reduction of grain yield [245,246,247,248,249,250]. These stresses can significantly alter the nutritional profile of the harvested seeds. As mentioned previously, the targeted micronutrients are either antioxidants or part of enzymes involved in various metabolic processes including electron transfer and oxidation reductions; thus, they protect cells from oxidative damage by quenching reactive oxygen species generated under environmental stresses [32,33,40,41,46,68,69]. Biofortified crops with a greater concentration of micronutrients can better withstand adverse environmental conditions and demonstrate improved adaptation under these conditions.

## 7. Conclusions

Micronutrients are essential for human growth and development, and their deficiency is a major concern that affects one in three people worldwide. Among various strategies, biofortification through plant breeding is considered the most economical and sustainable approach to tackle micronutrient deficiencies. This approach is universally accepted and has the potential to reach people living in relatively remote rural areas that have limited access to commercially marketed fortified foods. Further, it requires a one-time investment, and seeds can be multiplied across years by farmers at virtually zero marginal cost. In recent years, significant progress has been made with the release of several biofortified crop varieties that are helping to overcome micronutrient deficiencies in the target populations. Pulse crops are an important source of protein and energy, so improvement in their nutritional profile will significantly increase their consumption. Biofortification to improve the nutritional profile of pulse crops has gained momentum in the past decade. However, there are several challenges ahead that need to be addressed if the use of biofortified foods is to be successfully maximized.

## Figures and Tables

**Figure 1 plants-09-00073-f001:**
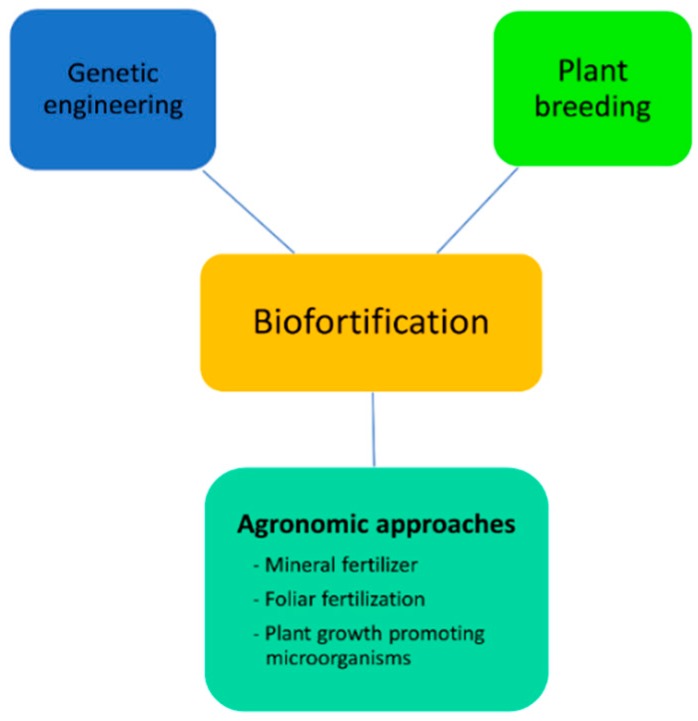
Different approaches of biofortification for improvement of nutritional profile.

**Table 1 plants-09-00073-t001:** Genetic variation and identified genomic regions and/or markers for the seed concentration of various micronutrients in pulse crops.

Micronutrient	Pulse Crop	Plant Material	Concentration Range (mg kg^−1^)	Genomic Region/Marker	Reference
Iron	Common bean	Genotype	34–89	7 QTLs	[169]
	Common bean	Genotype	35–92		[170]
	Common bean	DOR364 X G19833	40–85	13 QTLs	[171]
	Common bean	G14519 X G4825	36–97	5 QTLs	[172]
	Common bean	G21242 X G21078	28–95	6 QTLs	[173]
	Common bean	Genotype	30–110		[175]
	Lentil	Genotype	41–109		[174]
	Lentil	ILL 8006 X CDC Milestone	37–176	21 QTLs	[24]
	Lentil	Genotype	41–102	9 SNPs	[25]
	Lentil	Genotype	69–86		[28]
	Chickpea	Genotype	48–57		[28]
	Lentil	Genotype	76–100		[11]
	Pea	Genotype	48–58		[11]
	Common bean	Genotype	58–81		[11]
	Chickpea	Genotype	49–56		[11]
	Chickpea	Genotype	36–86	4 SNPs	[12]
	Chickpea	ICC 4958 X ICC 8261	40–67	6 QTLs	[23]
	Chickpea	Genotype	40–91	10 SNPs	[23]
	Mungbean	Genotype	35–87		[176]
	Pea	Genotype	26–94	9 SNPs	[13]
	Pea	PI 648006 X PI 357292	37–62	5 QTLs	[26]
	Pea	Orb X CDC Striker	26–49	4 QTLs	[27]
	Pea	Carerra X CDC Striker	34–67	6 QTLs	[27]
	Pea	Genotype	29–91	3 SNPs	[29]
Zinc	Common bean	Genotype	21–54	11 QTLs	[169]
	Common bean	Genotype	21–60		[170]
	Common bean	DOR364 X G19833	18–42	13 QTLs	[171]
	Common bean	G14519 X G4825	17–49	8 QTLs	[172]
	Common bean	G21242 X G21078	17–57	3 QTLs	[173]
	Common bean	Genotype	25–60		[175]
	Lentil	Genotype	22–77		[174]
	Lentil	Genotype	23–54	12 SNPs	[25]
	Lentil	Genotype	46–55		[28]
	Chickpea		35–43		[28]
	Lentil	Genotype	37–51		[11]
	Pea	Genotype	27–34		[11]
	Common bean	Genotype	25–33		[11]
	Chickpea	Genotype	21–28		[11]
	Chickpea	Genotype	19–62	5 SNPs	[12]
	Chickpea	ICC 4958 X ICC 8261	28–48	5 QTLs	[23]
	Chickpea	Genotype	27–62	10 SNPs	[23]
	Mungbean	Genotype	21–62		[176]
	Pea	Genotype	14–93	2 SNPs	[13]
	Pea	PI 648006 X PI 357292	31–62	5 QTLs	[26]
	Pea	Orb X CDC Striker	25–34	4 QTLs	[27]
	Pea	Carerra X CDC Striker	17–41	6 QTLs	[27]
	Pea	Genotype	13–51	7 SNP	[29]
Selenium	Lentil	Genotype	0.3–2.6		[180]
	Lentil	Genotype	0.01–0.3		[181]
	Lentil	Genotype	0.4–0.5		[28]
	Chickpea	Genotype	0.3–0.4		[28]
	Lentil	Genotype	0.9–1.6		[11]
	Pea	Genotype	0.4–0.5		[11]
	Common bean	Genotype	0.4–0.5		[11]
	Chickpea	Genotype	0.6–0.9		[11]
	Mungbean	Genotype	0.2–0.9		[176]
	Pea	Genotype	0.03–1.8		[182]
	Pea	Genotype	0.08–5.5		[13]
	Pea	Orb X CDC Striker	0.3–2.2	3 QTLs	[27]
	Pea	Carerra X CDC Striker	0.1–6.8	6 QTLs	[27]
	Pea	Genotype	0.1– 8.7	44 SNPs	[29]
Carotenoids	Chickpea	Not available		5 QTLs	[183]
	Chickpea	Genotype	311–880		[184]
	Chickpea	Genotype	11–19		[185]
	Chickpea	Genotype	9–31		[186]
	Chickpea	Genotype	22–44		[187]
	Chickpea	CDC Jade X CDC Frontier	15–58	8 QTLs	[188]
	Chickpea	Cory X CDC Jade	2–78	5 QTLs	[188]
	Chickpea	ICC4475 X CDC Jade	22–84	5 QTLs	[188]
	Pea	Genotype	7–23		[185]
	Pea	Genotype	6–27		[186]
Folates	Pea	Genotype	0.6		[189]
	Pea	Genotype	0.3–0.7		[190]
	Common bean	Genotype	1.4–1.6		[190]
	Lentil	Genotype	1.5–2.0		[190]
	Pea	Genotype	0.5		[87]
	Lentil	Genotype	0.7		[87]
	Faba bean	Genotype	1.0		[87]
	Chickpea	Genotype	1.5		[87]
	Chickpea	Genotype	2.7		[191]
	Common bean	Genotype	1.1–1.6		[191]
	Lentil	Genotype	1.1–1.5		[191]
	Pea	Genotype	0.1–0.2		[191]
	Pea	Genotype	0.4–2.0		[192]
	Chickpea	Genotype	0.4–1.2		[192]
	Lentil	Genotype	2.2–2.9		[192]
	Chickpea	Genotype	3.5–5.9		[15]
	Common bean	Genotype	1.6–2.3		[15]
	Lentil	Genotype	1.4–1.8		[15]
	Pea	Genotype	0.2–0.3		[15]
	Chickpea	Genotype	4.0–4.3		[193]
	Common bean	Genotype	2.4–3.0		[193]
	Lentil	Genotype	1.2–1.6		[193]
	Pea	Genotype	0.1–0.2		[193]
	Lentil	Genotype	1.7–5.0		[194]
	Pea	Genotype	0.1–0.6	31 SNPs	[30]

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
