# Peer review of "Biofortification of Pulse Crops: Status and Future Perspectives"

_plants, 2020, doi:10.3390/plants9010073_

Round 1
Reviewer 1 Report
Please receive my review to the manuscript entitled “Biofortification of Pulse Crops: Status and Future 2 Perspectives”. The review contains useful information for the scientific community on pulses biofortification and their approaches to improve their nutritional profiles. However, the review needs to add a component on the challenging environment factor such as high heat and drought and diseases on the breeding/genetic engineering selection/products (cultivars). How to integrate the environmental stress factors on the nutritional qualities, especially micronutrients content and profile. The review is well written and comprehensive and can be accepted with minor revision.
Comments;
-Some references are not consistent: for example, some journals are spell out and some are abbreviated. For example, see below.
Welch, R.M. Breeding strategies for biofortified staple plant foods to reduce micronutrient malnutrition globally. J Nutr. 2002, 132, 495S-499S. Patterson, C.A.; Maskus, H.; Dupasquier, C. Pulse crops for health. Cereals Foods World 2009, 54, 671 108-113. Khazaei, H.; Podder, R.; Caron, C.T.; Kundu, S.S.; Diapari, M.; Vandenberg, A.; Bett, K.E. Marker-trait association analysis of iron and zinc concentration in lentil (Lens culinaris Medik.) seeds. Plant Genome 2017, 10, doi: 10.3835/plantgenome2017.02.0007.
Author Response
Note that each of our responses is preceded by TW.
Reviewer 1:
Comments and Suggestions for Authors
Please receive my review to the manuscript entitled “Biofortification of Pulse Crops: Status and Future 2 Perspectives”. The review contains useful information for the scientific community on pulses biofortification and their approaches to improve their nutritional profiles.
- However, the review needs to add a component on the challenging environment factor such as high heat and drought and diseases on the breeding/genetic engineering selection/products (cultivars). How to integrate the environmental stress factors on the nutritional qualities, especially micronutrients content and profile.
TW: The following sentences have been added:
“In pulse crops, growth and productivity are affected by various abiotic and biotic stresses which can result in significant reduction of grain yield [240-245]. These stresses can significantly alter the nutritional profile of the harvested seeds. As mentioned previously, the targeted micronutrients are either antioxidants or a part of enzymes involved in various metabolic processes including electron transfer and oxidation-reductions, and thus protect cells from oxidative damage by quenching reactive oxygen species generated under environmental stresses [32, 33, 40, 41, 46, 68, 69]. Biofortified crops with greater concentration of micronutrients can better withstand adverse environmental conditions and improved adaptation in these conditions” (P17, P662-669).
The relevant references have been cited in the references section.
- The review is well written and comprehensive and can be accepted with minor revision.
Comments; Some references are not consistent: for example, some journals are spell out and some are abbreviated. For example, see below.
Welch, R.M. Breeding strategies for biofortified staple plant foods to reduce micronutrient malnutrition globally. J Nutr. 2002, 132, 495S-499S.
Patterson, C.A.; Maskus, H.; Dupasquier, C. Pulse crops for health. Cereals Foods World 2009, 54, 671 108-113.
Khazaei, H.; Podder, R.; Caron, C.T.; Kundu, S.S.; Diapari, M.; Vandenberg, A.; Bett, K.E. Marker-trait association analysis of iron and zinc concentration in lentil (Lens culinaris Medik.) seeds. Plant Genome 2017, 10, doi: 10.3835/plantgenome2017.02.0007.
TW: References have been edited for consistency.
Reviewer 2 Report
Issues that need to be addressed are listed below:
lines 8-9 | The sentence should be rephrased and include the “Genetic engineering approaches” as it is indicated in Figure 1 of the manuscript. line 35 | Several strategies both through traditional breeding and genetic engineering have been deployed to address the issue of micronutrient deficiency. A summary for such approaches in rice can be found in doi: 10.3389/fpls.2015.00121 line 42 | Add reference for the “slowly digestible carbohydrates" lines 163-164 | A reference is required for the statement. lines 254-255 | In rice, the national levels for Fe and Zn were attained under field settings (doi: 10.1038/srep19792). A recent summary of transgenic approaches for Fe and Zn in rice can be found at doi: 10.1007/978-3-319-95354-0_1. lines 576-577 | The sentence needs to be rephrased. What about the biotic/abiotic stimuli? A reference needs to be added. lines 598-600 | reference for the statement? lines 630-633 | Please refer to doi: 10.1007/978-3-319-95354-0_1 for retention of zinc after cooking for biofortified varieties produced either through traditional breeding or genetic engineering approaches.Author Response
Note that each of our responses is preceded by TW.
Reviewer 2
Comments and Suggestions for Authors
Issues that need to be addressed are listed below:
- lines 8-9 | The sentence should be rephrased and include the “Genetic engineering approaches” as it is indicated in Figure 1 of the manuscript.
TW: This review is mainly focused on biofortification through conventional plant breeding and genomic approaches. Therefore, in the abstract we have not mentioned genetic engineering and agronomic approaches, which are later referred to in Figure 1.
- line 35 | Several strategies both through traditional breeding and genetic engineering have been deployed to address the issue of micronutrient deficiency. A summary for such approaches in rice can be found in doi: 10.3389/fpls.2015.00121
TW: Strategies have been discussed later in this section:
“Biofortification, enriching the nutritional quality of food crop using either conventional plant breeding or modern biotechnology is a balanced approach to overcome mineral deficiencies [19-22]” (P2, L47-49).
- line 42 | Add reference for the “slowly digestible carbohydrates"
TW: The references (8, 16) have been moved towards the end of the sentence. Ref. 9 (Patterson et al. 2009) has been added (P2, L42).
- lines 163-164 | A reference is required for the statement.
TW: Reference 93 (White and Broadley 2009) and http://www.fao.org/agriculture/crops/ have been added (P5, L171-172).
- lines 254-255 | In rice, the national levels for Fe and Zn were attained under field settings (doi: 10.1038/srep19792). A recent summary of transgenic approaches for Fe and Zn in rice can be found at doi: 10.1007/978-3-319-95354-0_1.
TW: The following sentence has been added:
In rice, the national levels for Fe and Zn biofortification nutrition targets were attained under field settings in the Philippines and Colombia [144] (P7, L263-265).
- lines 576-577 | The sentence needs to be rephrased. What about the biotic/abiotic stimuli? A reference needs to be added.
TW: The sentence “For successful biofortification, high nutrient density must be combined with high yield and profitability. The efficacy of the crop must be demonstrated, and these crops must be adopted by farmers and consumed by the target population [21]” has been replaced with “Greater micronutrient density with high yield are prerequisites for effective biofortification and these crops must be adopted by farmers and consumed by the target population [21]” (P16, L597-598).
In addition, the following sentences have been added:
“In pulse crops, growth and productivity are affected by various abiotic and biotic stresses which can result in significant reduction of grain yield [240-245]. These stresses can significantly alter the nutritional profile of the harvested seeds. As mentioned previously, the targeted micronutrients are either antioxidants or a part of enzymes involved in various metabolic processes including electron transfer and oxidation-reductions, and thus protect cells from oxidative damage by quenching reactive oxygen species generated under environmental stresses [32, 33, 40, 41, 46, 68, 69]. Biofortified crops with greater concentration of micronutrients can better withstand adverse environmental conditions and improved adaptation in these conditions” (P17, P662-669).
The relevant references have been cited in the references section.
- lines 598-600 | reference for the statement?
TW: Reference 93 (White and Broadley 2009) has been added (P16, 620).
- lines 630-633 | Please refer to doi: 10.1007/978-3-319-95354-0_1 for retention of zinc after cooking for biofortified varieties produced either through traditional breeding or genetic engineering approaches.
TW: The following sentence has been added:
“Retention of zinc content after cooking in biofortified rice varieties produced either through traditional breeding or genetic engineering approaches has been discussed in detail by Tsakirpaloglou et al. [238]” (P17, L654-656).
Reviewer 3 Report
The review of “Biofortification of Pulse Crops: Status and Future Perspective” for Plants MDPI.
The topic of manuscript fits within the scope of the journal and results can be considered of interest in order to achieve an integrated exploitation of biofortification to improve the nutritional profile of pulse crops.
The type of article is a review and the authors have complied with all the requirements for such articles. The authors not only summarize the current state of research in field of biofortification, but also analyze and discuss it giving various data on research problems. For example in Section 6 “Challenges and Future Strategies for Biofortification” the role of polyphenols as nutrients/antinutrients is well discussed.
The manuscript is well prepared, nicely organized and written.
I have just a few remarks, which I give under author’s consideration:
1) In section 3.4.2. Genetic Engineering added some sentences about new gene editing methods based on TALENs and CRISPR/Cas9 used also for biofortification.
2) In Section 2 Key Micronutrients please add the RDA-values for Iron, Zinc, Selenium and Iodine
Author Response
Note that each of our responses is preceded by TW.
Reviewer 3
Comments and Suggestions for Authors
The review of “Biofortification of Pulse Crops: Status and Future Perspective” for Plants MDPI.
The topic of manuscript fits within the scope of the journal and results can be considered of interest in order to achieve an integrated exploitation of biofortification to improve the nutritional profile of pulse crops.
The type of article is a review and the authors have complied with all the requirements for such articles. The authors not only summarize the current state of research in field of biofortification, but also analyze and discuss it giving various data on research problems. For example in Section 6 “Challenges and Future Strategies for Biofortification” the role of polyphenols as nutrients/antinutrients is well discussed.
The manuscript is well prepared, nicely organized and written.
I have just a few remarks, which I give under author’s consideration:
- 1) In section 3.4.2. Genetic Engineering added some sentences about new gene editing methods based on TALENs and CRISPR/Cas9 used also for biofortification.
TW: The following sentences have been added:
“In recent years, targeted gene editing technologies using artificial nucleases, zinc finger nucleases (ZFNs), transcription activator-like effector nucleases (TALENs), and clustered regularly interspaced short palindromic repeat (CRISPR)/CRISPR-associated protein 9 (Cas9) system (CRISPR/Cas9) have given rise to the possibility to modify genes of interest precisely, and thus have potential application in crop improvement [152, 153]. These technologies have been used in various crops including rice [154, 155], wheat [156], and tomato [157]. Recently, CRISPR/Cas9 and TALENs technologies were used to generate mutant lines for genes involved in small RNA processing of Glycine max and Medicago truncatula [158]. Similarly, CRISPR/Cas9-mediated genome editing technology was used in cowpea to successfully disrupt symbiotic nitrogen fixation (SNF) gene activation [159]. These findings pave the way for applicability of use of gene editing technologies for various traits of interest in legumes” (P7, L282-292).
The relevant references have been cited in the references section.
2) In Section 2 Key Micronutrients please add the RDA-values for Iron, Zinc, Selenium and Iodine
TW: The following sentences have been added:
“According to the Food and Nutrition Board of the Institute of Medicine, National Academy of Sciences, the RDA of Fe is 8 mg/day for adult males and 18 mg/day for females (https://ods.od.nih.gov/Health_Information/Dietary_Reference_Intakes.aspx)” (P2, L63-65).
“The RDA for Zn is 11 mg/day for adult males and 8 mg/day for adult females (https://ods.od.nih.gov/Health_Information/Dietary_Reference_Intakes.aspx)” (P2, L75-76).
“The RDI for Se is 55 μg/day for both males and females (https://ods.od.nih.gov/Health_Information/Dietary_Reference _Intakes .aspx)” (P2-3, L82-84).
“According to the Food and Nutrition Board, Institute of Medicine, the RDI for I is 150 μg/day for both adult males and females (https://ods.od.nih.gov/Health_Information/Dietary_Reference _Intakes .aspx)” (P3, L88-90).